# 'Obstetricians' perceptions of midwifery-led care in Bangladesh – A qualitative study

Fflur Dafis[1,2]*, Abdul Halim[3], Abu Sayeed Md. Abdullah[3], Sumaiya Afrose Khan Atina[3], Terry Kana[4]

1 Department of International Public Health, Liverpool School of Tropical Medicine, Liverpool, United Kingdom, 2 School of Medicine, University of Liverpool, Liverpool, United Kingdom, 3 Reproductive and Child Health Department, Centre for Injury Prevention and Research Bangladesh, Dhaka, Bangladesh, 4 Faculty of Education, Liverpool School of Tropical Medicine, Liverpool, United Kingdom

* hlfdafis@liverpool.ac.uk

## Abstract

The midwife as a separate and autonomous profession is a relatively new concept in Bangladesh; as a result, its integration into the maternal healthcare system is yet to be determined. Obstetricians are key stakeholders in maternal health, and understanding how they perceive midwifery-led care can provide valuable insight into the acceptance of the model, and the profession itself. Data was collected from 13 semi-structured interviews conducted with obstetricians working in the Dhaka division, Bangladesh, in July 2023. Thematic analysis was then carried out on the transcripts. Four main themes were identified. 1. Diverse understanding of the role of a midwife; There was a general lack of understanding of the role of a midwife, and confusion surrounding their scope of practice. 2. Perceived benefits of midwifery; Obstetricians felt as though midwives can help decrease their workload, and that they are specialised practitioners who can help improve access and equity of healthcare. 3. Factors restricting midwives' professional autonomy; There was an evident lack of trust from obstetricians of midwives' competency and their education, and their acceptance and integration into the healthcare system are weak. 4. Strengthening future midwifery; Obstetricians suggested a need for improvements in midwifery education, as well as government promotion of midwives. The lack of awareness of the role and potential of midwives limits their professional autonomy. There is a need to increase awareness of other healthcare professionals, as well as the public, of the benefits of midwives and the midwifery-led model of care. Improved regulation is also needed, particularly in the private health sector and in the implementation of midwifery education.

**Data availability statement:** The subject of the interviews were potentially sensitive, if participants held strong views they may be identifiable to their colleagues, and making these transcripts freely available may compromise the anonymity of the participants. Participants did not consent to their interview transcripts being made fully available, and this was not approved by either ethical committee (LSTM or CIPRB). Additional queries regarding data availability can be directed to the LSTM ethics committee (lstmrec@lstmed.ac.uk).

**Funding:** The author(s) received no specific funding for this work.

**Competing interests:** The authors have declared that no competing interests exist.

## Introduction

Bangladesh has been working hard to achieve the Sustainable Development Goals (SDGs) set out by the United Nations. Target 3.1 is to decrease the global Maternal Mortality Ratio (MMR) to less than 70 deaths per 100,000 live births [1]. Bangladesh has made impressive progress with this indicator, decreasing to 176 deaths per 100,000 live births in 2013, from 574 in 1990 [2]. However, the rate has plateaued recently, and was estimated to be at around 123 per 100,000 live births in 2020 [3]. Methun, Haq, and Uddin attribute this plateau to the country's deficient maternal healthcare coverage [4].

Midwifery-led care (MLC) is a holistic model of care, where the midwife is the primary healthcare professional (HCP), responsible for all the care provided during a woman's pregnancy and childbirth. Core to this model is the promotion of birth as a normal physiological process and the avoidance of unnecessary interventions [5]. However, the medicalisation of birth means that an obstetric-led model is often favoured in hospitals [6]. This model is associated with higher numbers of interventions and less continuity of care when compared with a midwifery-led model [7].

To the public and other health care professionals, MLC is a relatively new concept in Bangladesh [8]. Professional midwifery was only introduced in the last decade, with the first cohort of diploma midwives graduating in 2015 [9]. The International Confederation of Midwives (ICM) [5] defines a midwife as an individual who; has undertaken a midwifery training programme, in line with the ICM "Essential Competencies for Basic Midwifery Practice" and the ICM "Global Standards for Midwifery Education", is recognised in the country they are located, and has the appropriate qualifications and competency to practise midwifery.

To become a midwife in Bangladesh, individuals must study a three year diploma course, the curriculum of which aligns with the ICM competencies [8,10]. 61% of the institutions required students to be able to write in English, and 32% required them to be able to write in Bangla [11]. Aspiring midwives are also required to achieve a Higher Secondary School Certificate [12]. However, according to a study by Bogren et al., only 34% of the nursing institutes that provide midwifery education had a transparent admissions policy; therefore it is unclear of the criteria required by prospective midwifery students [11].

Studies have found that internationally standardised midwives can provide up to 87% of essential care for mothers and their newborns [13]. However, in Bangladesh, there are complex social, professional, and economic barriers that hinder midwives' ability to reach their potential. Amongst these are the country's patriarchal and hierarchical society, a lack of professional autonomy for midwives, and irregular or inadequate salaries [14,15].

The introduction of midwifery in Bangladesh has been well documented in literature [8,9]. Studies have been undertaken that investigate the barriers and enablers of midwifery, explore different stakeholders' perspectives, and measure outcomes. However, there is a lack of research into obstetricians' perspectives. This study addresses this crucial gap in the literature. The position of

obstetricians at the pinnacle of the medical hierarchy makes them important stakeholders in Bangladesh's maternal healthcare. Therefore, understanding how they view MLC, their understanding of midwifery as a profession, and the barriers and enablers they identify are key to articulating the position of midwifery in Bangladesh today. By comparing the results of this study with previous literature, we can assess the continuity of perceptions of MLC. This will also allow us to analyse the differences between obstetricians' views towards midwifery, with those of other stakeholders. We hope that this study will be a valuable contribution to the existing literature, and will help inform policy and practice.

The research questions of this study can be seen in Table 1.

## Methodology

This was a qualitative study [16], with data generated from 13 in-depth interviews with obstetricians. 'Obstetricians' in this study included consultants, professors, registrars, or medical officers, specialising in obstetrics.

### Ethics statement

This study received ethical approval from both LSTM and CIPRB. Participants were informed that interviews were confidential, their participation was voluntary, and they could stop the interview at any time, or withdraw from the study up until the point of transcription. Oral and written informed consent was obtained from all participants (see S1 Text Participant Information Sheet and Consent Form).

Obstetricians who wished to participate in the study were required to take time out of their working day to be interviewed, meaning that they were taken away from their clinical or educational duties, for research that would not directly benefit them or the mothers and patients in their care. There was a risk of data being compromised if any equipment was lost or stolen, therefore all equipment was password-protected to help mitigate this risk, and recordings were transcribed and deleted within 48 hours. There is also a risk regarding the confidentiality of participants; if they hold strong views, they may be identifiable to their colleagues.

### Setting

Bangladesh, with a population of 169,356,251, is the 8th most populated country in the world [17]. Since 1980, there has been an impressive decline in poverty, from 41.9% to 13.5% in 2016 [18]. In 2015, they attained lower-middle-income status, and by 2031, aim to reach upper- middle income status [18].

According to the latest DHS survey, the caesarean section rate of live births in Bangladesh was 45% [19]. This is compared to the estimated global average of around 21% [20].

The DHS also states that 56% of births were assisted by a qualified doctor, 19% by untrained traditional birth attendants, and 13% by a nurse, midwife, or paramedic [19]. These statistics demonstrate a clear lack of midwifery-specific data.

A report by WHO states that there are an estimated 67 midwives per 10,000 population, and 317 nurses per 10,000 population. Of these 67 midwives per 10,000, 63 worked in the urban setting, and only 4 worked rurally. It

**Table 1. Research Questions.**

| Research questions |
| --- |
| How do obstetricians perceive midwifery-led care in Bangladesh? |
| What do obstetricians understand as the role of a midwife? |
| What do obstetricians identify as barriers and enablers of midwifery-led care? |

also states that there are 635 medical doctors per 10,000, but does not state the number of obstetricians specifically. [21]

## Sampling strategy

Initially, the sampling method was maximum variation purposive sampling. However, due to the difficulties of arranging interviews ahead of time, purposive criterion sampling was used based on the inclusion criteria. This allowed us to identify relevant and available participants who would be able to contribute to the discussion. Obstetricians in Bangladesh are very busy; therefore, this method was the most appropriate and feasible. It was also the most suitable to maximise the use of the limited time available.

The aim was to interview 10–15 obstetricians. Participants continued to be invited for interviews until no new themes emerged and saturation was reached. We deemed this to be the case after the 13th interview. The heterogeneity of the participants meant that saturation was easier to achieve. Time and financial constraints also guided the initial sample size range.

## Data collection

For this project, we worked with the Centre for Injury Prevention and Research Bangladesh (CIPRB). Senior staff from the organisation helped facilitate the study, and a research assistant helped with arranging interviews. Data was collected between the 9th and 20th of July 2023. Interviews mainly took place in two private hospitals in the Dhaka division, one in Dhaka City and one outside the city. These hospitals were chosen as they had an affiliation with CIPRB. Permission was gained by the medical directors of each hospital to conduct the interviews. Participants were recruited face to face and invited according to the inclusion criteria (see Table 2).

Interviews were conducted in private rooms in the hospitals, in the presence of a research assistant from CIPRB, who was able to help with communication issues and translate certain words to and from Bangla when occasionally necessary. Interviews were conducted in English and semi-structured according to the topic guide (see S2 Text). Questions ranged from general ('Could you describe how maternity care is provided in Bangladesh?'), to specific ('Do you envisage a role for yourself in facilitating or supporting the roll out of midwifery-led care?').

The topic guide was informed by previous literature and key informant interviews. The interviewer aimed to gain an understanding of the personal opinions of the participants, prompting for examples and development of points when necessary. Interviews were recorded electronically. The majority of interviews lasted between 30–40 minutes. Although some interviews lasted less than this, even shorter interviews provided recurring themes. This was influenced by how busy the participants were on that day and how comfortable they felt speaking in English.

Interviews were conducted in English, as all obstetricians receive their medical training in English and were therefore all competent and comfortable to interview in English. Conducting interviews in English also contributed to the reliability of the research; having the lead author conduct the interviews, analyse the transcripts, and formulate the themes allowed for continuity.

**Table 2. Inclusion Criteria.**

| Inclusion Criteria |
| --- |
| - Obstetricians |
| - Work in clinical practice, education or research |
| - Agree to interview in English |
| - Based in Bangladesh |

## Analysis

Interviews were recorded on a password-protected device and transcribed in detail, including verbal nuances and notes of body language, within 48 hours of the interview taking place. Transcripts were then rechecked before recordings were deleted. NVivo software was used to store, organise, and analyse the data. Thematic analysis was used as it allowed for the extraction and interpretation of fundamental patterns, leading to the development of rich themes [22]. Transcripts were read several times to promote familiarisation. Using NVivo software, transcriptions were coded to fragment the data, based on specific thoughts and ideas. Related codes were grouped together to form subthemes. The relationships between these subthemes were further analysed to determine how they addressed the research questions, leading to the formation of broader themes. The coding process was guided by the data itself and by our research questions. Thematic analysis is subject to the researchers' judgement; to mitigate this risk of bias, themes were discussed with supervisors in Bangladesh and in Liverpool, as well as with the research assistant. These discussions helped refine and finalise the themes. The naturalistic paradigm was core to this research. Perceptions are subjective; therefore, understanding their complexity and context is essential to fulfil the research aim [23].

## Trustworthiness

Participant checking was used to promote trustworthiness. When uncertain, the interviewer would repeat back to the participant and ensure that they were interpreting it correctly. The research assistant helped promote trustworthiness by translating words or phrases into Bangla if participants did not understand the question. However, this may have meant that some data was lost in translation.

Asking open questions and avoiding leading questions was important to promote the free expression of participants, and ensure credibility. Seeking feedback on the topic guide, and adapting prompts as interviews progressed also furthered the credibility of the study. To strengthen the dependability of the study, the same topic guide was used for all interviews. Having the same person conduct the interviews, transcribe, and analyse the data allowed for continuity, but limited the opportunity for triangulation and likely increased researcher bias. To promote confirmability, a reflective journal was kept by the researcher to acknowledge and minimise personal bias, and to reflect on their positionality throughout. Providing the details of participant characteristics and the recruitment process, as well as including the interview guide and analysis steps enables the transferability of this study to other contexts.

## Results

Participants varied in terms of their demographics and experience; see Table 3 for participants' characteristics.

There were four main themes identified from the data: diverse understanding of midwives' scope of practice; perceived benefits of midwifery; factors restricting midwives' professional autonomy; and strengthening future midwifery (see Table 4).

**Table 3. Participant Characteristics.**

| Characteristics | | Number of participants |
|---|---|---|
| *Sex* | Female | 11 |
| | Male | 2 |
| *Job Title* | Professor | 2 |
| | Consultant | 8 |
| | Registrar | 2 |
| | Senior Medical officer | 1 |
| *Public/Private Sector experience* | Private | 6 |
| | Public and Private | 7 |

**Table 4. Themes and Subthemes.**

| Themes | Subthemes |
|---|---|
| **Diverse understanding of midwives' scope of practice** | Support and monitor uncomplicated pregnancies<br>Grouped together with nurses |
| **Perceived benefits of midwifery** | Improving access and equity<br>Decreased workload<br>Specialised practitioners |
| **Factors restricting midwives' professional autonomy** | Integration and acceptance of midwives into the healthcare system<br>Lack of trust in midwives' competency<br>Lack of trust in midwives' clinical education |
| **Strengthening future midwifery** | Government promotion<br>Curriculum adaptation<br>Increased clinical experience |

### Diverse understanding of midwives' scope of practice

The perceived role of a midwife and their scope of practice varied between participants, especially between those working in Dhaka city and outside the city. In the city, the obstetricians described the role of a midwife as a predominantly supportive role, where they monitor and counsel mothers with uncomplicated pregnancies.

> *"They can [conduct] deliveries, normal cases, easy, normal deliveries. But when they face complication, that is prolonged labour, obstetric labour, malpresentation, they refer."*
>
> *– Consultant, female, private experience only*

Obstetricians outside of Dhaka city had noticeably less awareness of the role of a midwife. Several participants stated that they were aware that there were midwives employed at the hospital, but were unable to differentiate between the nurses and the midwives. Perceiving the role of midwives and nurses as the same was not exhaustive; however, it was recurring. It was also evident that midwives working outside of Dhaka city had different roles compared to those in the city, with midwives outside of the city having more administrative responsibilities and less clinical autonomy. Participants stated that vaginal births were mainly conducted by interns and medical officers. Midwives and nurses were more likely to conduct births at night when there were fewer doctors available.

> *"In our hospital we cannot differentiate between (a) nurse and (a) midwife."*
>
> *– Registrar, female, private experience only*

Examples given for the day-to-day role of midwives were; monitoring and measuring the progress of mothers in labour, being the first to identify complications, and counselling mothers on foods and vaccines. Several participants stated that in uncomplicated cases, with no risk factors, midwives can conduct normal vaginal births. Examples of restrictions on the scope of practice of midwives were: any complicated pregnancies, performing an episiotomy, repairs of tears etc, assisted vaginal birth, and placental removal. There was a heavy emphasis on referring to the doctor if there were any complications, and that midwives were under constant supervision by the doctors.

> *"… normally they [midwives] are not doing the delivery. They are doing when nobody is present. If babies coming out … then they do, especially in evening or night, due to scarcity of doctors."*
>
> *– Assistant professor, female, private experience only*

A few participants spoke about how, compared to other countries, such as the UK, midwives in Bangladesh are less independent and have less autonomy, but that efforts are being made to improve this. When asked about the lead health cadre for maternal care, the majority identified doctors as the lead, but some participants expressed that this depended on the facility, explaining that in the community the midwife may be seen as the lead.

### Perceived benefits of midwifery

**Decreased workload.** One of the main subthemes identified was the perception of a decreased workload for obstetricians as a result of MLC, subsequently relieving their stress. A few participants stated that midwives were essential to increase the vaginal birth rate in Bangladesh's facilities. They explained that to facilitate safe vaginal birth, which can be a long process, additional personnel are needed. Midwives are also essential to be able to provide 24-hour care. They also described how having midwives to monitor and reassure women allowed doctors to use their time more efficiently, such as attending to high-risk mothers. Some participants described that having midwives provide primary healthcare to mothers in the community, whereby mothers were only referred to the obstetricians when necessary, also significantly decreases their workload.

*"…doctors are very busy. Normal delivery is a long procedure and takes a long time. If we want to say that we increase normal delivery rate, safe delivery rate, and we want supervision of our mothers with respectful maternity care…if we engage particular groups like midwives…it's possible to take care of our safe deliveries."*

*- Consultant, female, private experience only*

One participant spoke of a particularly reliable and skilled midwife working in their hospital; her competency and ability to work independently meant that she was trusted to monitor the mothers, and escalate to the obstetricians if needed. As a result, the obstetricians were able to assess, investigate, and manage more mothers.

*"In our hospital there is midwife…she is very skilled. She conducted the delivery, she can inform us about the progress of the labour very efficiently, so we can rely on her; we can see the other patients here, in labour room she is monitoring the patient and informing us over telephone. If she faces any difficulties, she can inform us…she is very skilled. This changes our job. Because we can give more time here."*

*- Consultant, female, public and private experience*

**Specialised practitioners.** Some participants appreciated that midwives were a unique cadre, in terms of their specific training, especially as opposed to nurses who receive limited specific maternal training.

*"…they are very much dedicated…that is because they work only in narrow sector, they are very much well [knowledgeable] about this"*

*- Consultant, female, public and private experience*

One participant stated that midwives in the community are especially skilled, due to their experience of facilitating births independently. This participant also appreciated that having competent midwives, who can practise autonomously, could have a huge impact on maternal health in Bangladesh. Several participants appreciated the importance of midwives in rural communities, where there are no or few doctors.

*"[In] rural area, midwives delivery individually. By practice they are skilled."*

*- Senior consultant, female, public and private experience*

Participants also identified that midwives promoted positive birth practices, such as breastfeeding, and skin-to-skin contact. They also described midwives as being very friendly, with the ability to build a good rapport with mothers, and especially good at supporting them during labour.

*"They're good at learning about normal delivery and other things. They are so much friendly. And they have a knowledge about how to care for the mothers."*

- Consultant, female, private experience only

**Improving access and equity.** Participants described how difficult it can be for Bangladeshi women to physically access medical care, and how providing midwives locally, in the community, makes maternal care more accessible to these women. One obstetrician described how posting midwives at the community level, where there are limited or no doctors, means that fewer mothers will need to be referred to hospitals. They emphasised that this was especially important for women in very remote, rural areas, with limited access to hospitals.

*"…doctors are not available at the grassroot level...So if we can establish increased number of [midwives], they can handle the situation. Patient need not refer. There may be riverside people, peripheral people. There is no communication, road is not available. So they can serve the people there, at least primary care. Normal delivery. So it is very important."*

*- Consultant, female, private experience only*

One specifically spoke about the role of midwifery in helping provide equitable care to the women of Bangladesh, illustrating how working alongside midwives allows obstetricians to spend more time with high risk mothers.

*"Suppose you [have] three patients, I allocate 5 minutes, 5 minutes, 5 minutes. She needs 1 minute, you need no minute…she needs half an hour. Now if I allocate everyone 5 minutes, I am doing an injustice to her...So this is why, are we doing too much or too [little]. So distribution of the risk involved matching with the scenario, that is very important."*

*- Professor, male, public and private experience*

This participant also described midwifery as cost-effective, stating that it allows for optimal use of the country's scarce resources.

*"People like us [obstetricians], country like us, where our limited resource. But we want to maximise this resource by involving the midwife".*

*- Professor, male, public and private experience*

**Factors restricting midwives' professional autonomy**

**Lack of trust in midwives' competency.** Many participants spoke of midwives being unable to identify complications, resulting in delayed escalation and referral. Obstetricians described how midwives' inability to diagnose prolonged labour, malpresentation, or other complications has previously led to late identification of mothers' complications by the doctors, and subsequently delayed management. Several participants stated that some midwives were not aware of the indicators for caesarean section, and were unable to identify which mothers should deliver normally.

*"They cannot understand when there is prolonged labour, malpresentation, malposition…They call for us late… There sometimes is mishap, where patient needs NICU submission, emergency c-section, but party is not ready for*

*caesarean section, they [think] that everything is okay, then what happens now, in the eleventh hour? They cannot understand all cases…"*

*- Consultant, female, private experience only*

Examples of delayed referral were given for midwives in the hospital and the community, with one participant stating that often, by the time the community midwife has identified a complication, it is too late, and the patient may die on the way to the hospital.

*"…in any difficult situation they refer. But in such a critical condition…sometimes that patient lose their life in ambulance."*

*- Senior consultant, female, public and private experience*

Some obstetricians spoke of midwives being able to identify complications, but not understanding the pathophysiology and, therefore, not understanding the necessary management. Others were critical of midwives' ability to identify any complications.

*"Even vulval haematoma they cannot identify"*

*- Senior consultant, female, public and private experience*

**Lack of trust in midwives' clinical education.** The education of midwives was discussed a lot in the interviews, both clinical and theoretical aspects. Participants mainly expressed their opinions on the lack of clinical experience that graduated midwives had, resulting in a lack of competency when qualified. Their lack of experience conducting normal vaginal births was a recurring perception. Some participants believed that this gap in their education was contributing to their inability to identify complications in a timely manner. Obstetricians spoke of the lack of clinical placement in their diplomas, stating that they learn in theory but lack practical skills. Claiming that their lack of clinical experience during their diploma leads to a need for more supervision while working. A few obstetricians referred to the curriculum, describing it as inadequate and in need of reform. Participants also spoke of the ineligibility of current midwifery educators.

*"But in our country, they are not posted during their training period, so how can they train to deliver individually [deliver on their own]? They only learn but not do. They [do] not do in practical scenario, so they are not efficient."*

*- Senior consultant, female, public and private experience*

**Integration and acceptance of midwives into the health care system.** Some obstetricians identified the novelty of the midwifery profession as a barrier, stating that midwifery care is not widely accepted in Bangladesh. Several obstetricians described midwifery as a foreign concept to the people of Bangladesh. One explained that this, coupled with the medicalisation of Bangladesh's health care system, sometimes causes hostility towards midwives, both from the public and medical professionals. One participant suggested that some doctors may be threatened by midwives and that there may be a sense of fear that midwives would be preferred to them and that they would take some of their work, leading to a lack of acceptance by obstetricians.

*"I think there will be some problems arising when both of them come [midwife and obstetrician]…If you think that doctor cannot do a proper delivery, and that midwife is good at delivery, then there will be a problem. If it is happening that senior is liking that midwife but she is not liking that doctor…this type of problem will arise"*

*- Assistant professor, female, private experience only*

This participant also referenced the hierarchical medical system in Bangladesh, describing how doctors are seen as superior to other healthcare professionals, and how this acts as a barrier to improving maternal care in Bangladesh.

*"Previously sisters [nurses] was a third class citizen, and nowadays they are second class citizen. And we are 1st class citizen… But I think…they [midwives] must come for Bangladesh."*

*- Assistant professor, female, private experience only*

**Strengthening future midwifery**

Participants had several suggestions on how MLC could be promoted in Bangladesh. Adaptation of the curriculum was suggested by a few participants, of both theoretical and clinical aspects. The need to focus on normal vaginal birth was emphasised. Many suggested that if midwifery students performed more vaginal births, they would be able to better identify complications. Significant emphasis was placed on the ability to identify complications.

*"…they should improve (the) curriculum about knowledge, and also practical (skills). They have to … perform more deliveries so they have experience, that (they can identify) complications like breech delivery, normal delivery prolonged, partograph showing there may be some problem so (they can) inform or call (the) doctor earlier."*

*- Consultant, female, private experience only*

Some participants expressed the need for more suitable midwifery educators. A few participants suggested that obstetricians should be involved in teaching midwives; others suggested that it should be a multidisciplinary approach, with both nurses and obstetricians involved. It was also suggested by one participant that obstetricians should act as mentors to midwifery students, as well as to qualified midwives.

*"[the] midwifery course. That should be dramatic change. Because obstetricians should teach them. Not nurses. Because that is not possible, obstetrician should teach…1 or 2 or 3 months is not enough. [They need] at least 6 months to one year training in labour room, is very much important."*

*- Senior consultant, female, public and private experience*

Obstetricians highlighted the need for better education facilities, more placements in public hospitals, and a safe working environment for midwives. One participant suggested that the government should promote the role of a midwife to educate the public on their work, and how they can help the mothers of Bangladesh.

*"Proper training, ensure her working area is safe, that is very important. Her working atmosphere should be safe… Working atmosphere, she should feel that she is comfortable at discharging her duties."*

*- Professor, male, public and private experience*

*"There should be more work from the government level, for publicity about their work.."*

*- Consultant, female, public and private experience*

## Discussion

The purpose of this study was to explore obstetricians' perceptions of MLC. By doing this, we hoped to gain insight into their understanding of midwifery, particularly how MLC is viewed by obstetricians, and what they perceived to be the barriers and enablers of this model of care.

**To identify obstetricians' understanding of the role of a midwife**

There was a general lack of understanding of the role and scope of practice of midwives, with obstetricians' interpretations differing from global definitions [5,24]. Their perceptions were especially different in terms of midwives' ability to provide comprehensive healthcare beyond pregnancy and childbirth. A clear disparity is highlighted in Bangladesh regarding midwives' provision of holistic sexual and reproductive health care, compared to that stated by the ICM. The majority of participants made no reference to the role of midwives in the context of health education, or the broader sexual health context, as stated by the ICM [25]. The International Confederation of Midwives [26] lists family planning, sexual health, health promotion, and many more services under the competencies of midwives; however, these were not identified by the obstetricians as being within midwives' scope of practice.

The disparities between perceptions may be influenced by hospital policy, and different exposure to professional midwives. The lack of differentiation between midwives and nurses also emphasises the lack of awareness of midwives' scope of practice, and the need for improved policy and national promotion of midwives. This supports findings by Byrskog, Akther [14], who stated a general lack of understanding of midwives' role, and by Bogren, Erlandsson [15], who reported a lack of awareness by midwives of their scope of practice and limitations. Lack of awareness and confusion regarding the role of a midwife has also been reported by Bangladesh's Ministry of Health and Family Welfare [2], emphasising that this perception is not unique to obstetricians, and is a known barrier.

**To explore obstetricians' understanding and experience of midwifery-led care**

The view that midwives help decrease the workload of obstetricians was heavily reported. However, due to the relatively small number of midwives who have been deployed to government and private health facilities, the transferability of this perception across Bangladesh is weak. This perception does however, demonstrate a level of appreciation for the MLC model. It also coincides with existing literature on the potential of midwives [27]. Obstetricians reporting improvements in practice also aligns with previous literature [28,29]. Participants also perceived that midwives would be able to help increase rates of normal vaginal birth. This is supported by existing literature surrounding the MLC model's promotion of birth as a normal physiological process [30,31]. However, it is important to recognise that the factors influencing Bangladesh's high caesarean section rates are multifactorial and complex. These range from medical factors such as lack of antenatal care, to non-medical factors, including socio-economic status and cultural preferences [32].

The best example of MLC was given when discussing midwives working in the community. However, in general, there was a lack of experience of internationally standardised MLC. It is important to consider that the majority of government-employed midwives work in Upazila Health Complexes (primary health care facilities) in the community, where there are no obstetricians, and only around 380 midwives are working in private facilities [33]. As a result, obstetricians have limited exposure to the potential of midwives and MLC. Although there were midwives employed at the hospitals involved in the study, the lack of policy surrounding the employment of midwives and their scope of practice may have contributed to the ambiguity in participants' awareness.

One significant finding from this study is the lack of professional autonomy of midwives working in hospitals. Obstetricians' lack of trust in their education and competency influences this. Fear of litigation is also a probable influence; participants highlighted the lack of legal protection for healthcare professionals in Bangladesh, which inevitably influences obstetrician decision-making and willingness to delegate, especially as their high position in the medical hierarchy means that they are often the ones who bear responsibility for adverse outcomes.

**To document the barriers and enablers of midwifery-led care identified by obstetricians**

There was a clear lack of trust in midwives' education expressed by obstetricians, specifically surrounding their clinical experience. This complements existing research detailing midwives' lack of clinical practice [11,15,28]. Participants' view of midwifery educators being unqualified and inexperienced has also been stated by Erlandsson, Byrskog [34].

Their lack of trust in midwives' competency was evident. General distrust in midwives' competency has been acknowledged by midwifery students in existing literature [15], as well as the low priority of the standard of care in the MLC model [30]. However, when discussing late referrals, there was a lack of recognition of other factors influencing the late presentation of critically ill mothers to the hospital. This lack of trust also contributes to midwives' low levels of professional autonomy. The lack of integration and acceptance of the MLC model into the existing healthcare system was also significant. This supports previous literature that discusses the patriarchal norms surrounding decision-making, as well as the hierarchical medical system [15,28]. This hierarchy, and the consequential power dynamics that exist between professions, limit midwives' ability to provide comprehensive maternal care and the subsequent benefits of this.

Obstetricians' suggestions to change the curriculum imply that their understanding of how the curriculum has been designed is low. Lack of awareness of the curriculum by clinical placement sites was also reported by Bogren et al [11]. The midwifery curriculum in Bangladesh was designed and recently updated, in line with ICM standards [35]. This lack of awareness of how midwives are trained also influences their lack of autonomy. Although this internationally standardised curriculum is ubiquitous in Bangladesh, the same cannot be said about its implementation. Nuruzzaman, Zapata [36] describe a lack of regulation of education programmes in the country, resulting in an inconsistency in how the curriculum is applied in practice.

Obstetricians expressed their view that midwifery educators were unsuitable and often unqualified. The inadequate training of midwifery educators, in terms of the clinical aspect of the course, has been highlighted by Khatun et al previously [28]. According to a systematic review of midwifery education in Bangladesh, although the midwifery curriculum is of a high standard, the lack of confidence and experience of midwifery educators acts as a barrier to the standard of student midwives' education [8].

The National Midwifery Policy of Bangladesh lists ensuring the majority of the midwifery faculty are midwives as one of its objectives [2], emphasising that this is a known barrier. Some suggested that obstetricians should be involved in the training of midwives. Previous studies have highlighted the benefits of having obstetricians mentor midwives, as their position in the medical hierarchy can be utilised to advocate for them [28]. Due to the novelty of the midwifery profession, there is a lack of experienced senior midwives in Bangladesh to mentor and supervise future generations; therefore, obstetricians are arguably the most suitable mentors for midwives in this context. However, this is contrary to the MLC model, and obstetricians would need support in understanding the scope of practice of midwives. Additionally, the feasibility of this is debatable, with obstetricians already under significant pressure. The potential benefit of this would be to promote trust and confidence in midwives, who are currently taught by nurses who potentially have limited experience of midwifery.

Government promotion of midwives was suggested by participants, which coincides with findings by Sangy, Duaso [37], who expressed that women need to have a level of understanding and trust in midwives to access MLC.

### Contrast with previous literature

Although there were many similarities between this study's results and existing literature, obstetricians did not refer to many of the main barriers and enablers of MLC identified in previous research. Obstetricians did not identify financial barriers or limited resources [14,15,28]. There was also a limited reference to sociocultural norms and traditional practices [14,15], as participants work in similar contexts, they often seemed to view this environment from an insiders perspective, making it difficult to perceive these norms objectively. The use of mentors to facilitate and promote MLC was also mentioned sparingly [28,29]. Obstetricians' interpretation of barriers and enablers was arguably narrower than what previous literature has identified, this may have been due to a lack of awareness of how these factors influence MLC, but may also be attributed to the topic guide. The naturalistic paradigm is important here, as it allows us to consider why participants may perceive barriers and enablers differently from others. Obstetricians' knowledge of these barriers and enablers is shaped by their individual experiences, which explains why they have slightly different perceptions from other cadres. Understanding this is essential to appreciating the complexity of MLC implementation.

## Strengths and Limitations

Qualitative research methods are key to this study. They are unique in their ability to explore complex topics in depth, allowing for profound understanding. By conducting in-depth interviews, we were able to generate rich, contextualised data. This is particularly important for this research, where understanding the nuances of the participants' perceptions is essential. Conducting these interviews face-to-face was imperative to contextualise the data. Observing the body language, facial expressions, and tone of voice of participants helped the interviewer understand their perceptions. However, it is important to appreciate that the interpretation of body language is influenced by each individual's cultural background, and may differ between the UK and Bangladesh. Focus group discussions could have added another dimension to this research, but due to logistical constraints, they were not feasible. Quantitative methods would not be suitable for this research as they would be limited in terms of depth and context of the data. Before the data collection stage of this research, a critical review of the existing literature was undertaken. This helped to provide context and structure for the topic guide.

Although over half of the participants had experience working in public hospitals, the majority had worked exclusively in the private sector for the last few years. Therefore, as the introduction of professional midwives was in the last decade, their experience of midwives in a public sector setting would be minimal. The lack of obstetricians with recent experience working in public hospitals is a significant limitation of this study. Therefore, although the results of this study are significant in terms of obstetricians working in private hospitals, the transferability of the results to government doctors may be weak. This may also be weakened due to the small sample size in comparison to the population of obstetricians in Bangladesh.

Although the participants could speak English, there was a difference in the participants' ability to understand questions and in their confidence to express themselves in English. Participants often interpreted interview questions differently. This was likely influenced by the fact that participants were speaking in their second language.

The positionality of the interviewer is likely to have influenced participants, due to them being a medical student from the UK. Although having a medical background appeared to be beneficial in some instances, it may have increased the risk of social desirability bias, especially as many participants were aware of the UK's midwifery care.

The research was potentially subject to negativity bias, a cognitive bias that suggests that negative experiences influence us much more than positive ones [38]. Additionally, prior to the data collection, two doctors had been arrested for the death of a mother and her baby, and protests were being held during the data collection stage, which increased pressure on the doctors working and may have increased bias. The affiliation of the hospitals involved in the research with CIPRB also increases selection bias.

## Recommendations

There is a need to strengthen policy and improve regulation surrounding midwifery, especially in the private sector, as private facilities account for 45% of births [19].

To promote more vaginal births, in line with the MLC model, facility-based audits need to be conducted to identify examples of good practice and areas of improvement. This requires a multi-disciplinary approach, which will need nurturing due to Bangladesh's highly medicalised approach to childbirth.

Investment in community-based behaviour change initiatives is also necessary to promote midwifery, and advocate for facility-based births. However, it is equally essential to increase investment in ensuring that timely referral systems are in place and effective.

There should be efforts made to increase the awareness of healthcare professionals, as well as the community, of the role of midwives and their benefits, to promote acceptance of the profession and facilitate a supportive environment where midwives are respected and able to practice autonomously. Previous studies have identified this as an issue, and have

also recommended it. This particularly needs to be disseminated to medical professionals, including obstetricians, paediatricians, and physicians working in facilities where maternal healthcare is provided.

There should be a drive on a national policy level, with professional guidelines aimed at all health care professionals, which outlines who leads the care for normal pregnancy, birth, and post-natal period, and to reinforce the referral system.

The introduction of midwifery mentors into healthcare facilities could promote advocacy for the profession and facilitate supervision for midwifery students and graduates. Initially, obstetricians could be trained as midwifery mentors, while midwifery graduates continue to develop clinical expertise in their roles.

Regarding the curriculum, discussions with the OGSB are needed to ensure student midwives have adequate opportunities for clinical experience, especially as it is likely that there will be competition within facilities between trainees of different cadres to attend births.

Further research should be undertaken on this topic by conducting focus group discussions with obstetricians and quantitative questionnaires, looking at both government and private doctors' perceptions. This will help facilitate the triangulation of results.

## Conclusion

This study demonstrates that there is a general disconnect between obstetricians' understanding of the role and scope of practice of Bangladeshi midwives, and international definitions. Although there was acceptance of the profession and a sense of gratitude that midwives would be able to decrease obstetricians' workload, recognition of their professional autonomy is low. Lack of trust in their competency and education is a significant driving factor of their low autonomy. This is especially the case for hospital midwives, where medically-led maternal healthcare is the norm. This study also suggests that a lack of awareness by obstetricians and the public of midwives' scope of practice is restricting their potential. MLC, which is well integrated into the healthcare system, and clear communication and understanding between HCPs, can help Bangladesh to optimise the midwifery profession, improve maternal and neonatal health outcomes, and reach SDG targets.

## Supporting information

**S1 Text. Participant information sheet.**
(DOCX)

**S2 Text. Topic guide.**
(DOCX)

**S1 Table. Analysis matrix.**
(DOCX)

**S1 Checklist. Inclusivity in global research questionnaire.**
(DOCX)

## Acknowledgments

We extend our sincerest gratitude to the obstetricians who participated in this study, and to the key informants for their vital insight. We also thank the teams at OGSB, UNFPA, and CIPRB who were invaluable to this study.

## Author contributions

**Conceptualization:** Terry Kana.

**Methodology:** Abdul Halim, Abu Sayeed Md. Abdullah, Sumaiya Afrose Khan Atina, Terry Kana.

**Project administration:** Sumaiya Afrose Khan Atina.

**Resources:** Abdul Halim, Abu Sayeed Md. Abdullah.

**Supervision:** Abdul Halim, Terry Kana.

**Writing – original draft:** Fflur Dafis.

**Writing – review & editing:** Fflur Dafis.

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
