## [Decision Letter · Decision Letter 0]

20 Dec 2024

PGPH-D-24-02552

‘A qualitative exploration of obstetricians’ perceptions of midwifery-led care in Bangladesh’

Dear Dr. Dafis,

Thank you for submitting your manuscript to PLOS Global Public Health. After careful consideration, we feel that it has merit but does not fully meet PLOS Global Public Health’s publication criteria as it currently stands. Therefore, we invite you to submit a revised version of the manuscript that addresses the points raised during the review process.

The manuscript has been assessed by two reviewers and their comments are available below. The reviewers have identified a number of major concerns about the manuscript, particularly with respect to details of the methods for the conduct and analysis of the interviews. Could you please carefully revise the manuscript to address the concerns raise?

We look forward to receiving your revised manuscript.

Kind regards,

Marianne Clemence

Staff Editor

Journal Requirements:

2. Please provide separate figure files in .tif or .eps format.

Additional Editor Comments (if provided):

Reviewers' comments:

Reviewer's Responses to Questions

**Comments to the Author**

1. Does this manuscript meet PLOS Global Public Health’s publication criteria?

Reviewer #1: Partly

Reviewer #2: Partly

2. Has the statistical analysis been performed appropriately and rigorously?

Reviewer #1: N/A

Reviewer #2: N/A

3. Have the authors made all data underlying the findings in their manuscript fully available (please refer to the Data Availability Statement at the start of the manuscript PDF file)?

Reviewer #1: No

Reviewer #2: No

4. Is the manuscript presented in an intelligible fashion and written in standard English?

Reviewer #1: Yes

Reviewer #2: Yes

Reviewer #1: Thank you for your important work. I have a few comments that the authors may consider:

Introduction lines 42-43: it is a bold statement without any supporting reference. I would suggest adding a reference or rephrasing the opening line.

The introduction has been well constructed, but could be organised better, so that the text flows from current situation, context, facilitators, barriers and their impact on the profession, ending with why obstetricians' perceptions and experiences mattered and formed the core for this piece of work. Right now a lot of back-and forth happens in the text which may be confusing to readers outside of the region.

Methodology: lines 82-84: I wonder whether data saturation was possible with just 13 "semi-structured" interviews. Do the authors mean in-depth interviews? If not, a paragraph is needed on how was data saturation identified and consented upon by all co-authors.

line 121 Table 2: Consenting to or not consenting to participate cannot be called an inclusion or exclusion criteria. If they did not consent, it implies they were eligible for inclusion - thus must have met inclusion criteria - but refused participation. Therefore, I am requesting deletion of this criterion from the table.

lines 124-132: interviews that lasted barely 10 min raise significant concerns in my opinion, in a qualitative study. how many interviews ended in less than 40 min? Even 40 min interviews could be considered just adequate in terms of allowing an in-depth/holistic exploration of issues discussed, especially looking at the thematic guide (Appendix A). Authors need to justify conducting interviews in English/translation into English during the interview itself, rather than employing local interviewers to conduct full interviews in Bangla, especially in the light that authors knew the obstetricians to be busy and, then taking the dual language interviews that would have further cut into the actual discussion time of 10-40 min interviews.

Lines 174-178: It will be great if you could add the approval certificate numbers from both the institutions.

Lines 184-188: Authors state that the recordings were deleted within 48 hours, after the transcription was complete. I would like to better understand how was the analysis completed without the recordings, considering the pressure put on certain words, tone, and associated voice inflections are all qualitative data, especially in emotive coding which is commonly used for perceptions and experiences studies. Sometimes the weight put by the participants on certain words changes the intention drastically (from serious reflection to a taunt or a patronizing remark for example). If the authors captured all the inflections via detailed transcription, this should be made clear.

Findings:

Overall, I request the authors to reflect on the interchangeable terms that inadvertently mix the midwife-led care philosophy with the medical model of care philosophy. For example: reflect on your positionality on birth/childbirth vs. delivery; women/mothers vs. patients and so on.

The manuscripts presents some very interesting findings, but for the most part, the quotes are too short and too few to establish that the findings are rooted in actual interview data. Consider giving longer quotes or several short ones to demonstrate how you reached to the conclusions that you reached, with all the rich context outlined in your text.

Discussion:

Lines 436-438: authors mention family planning services. Sorry if I missed it, but I did not see any reference to FP in the results section.

Limitations: I feel the authors should add the study strengths as well, since this is a critical evaluation of the research process which may help future researchers exploring midwifery research in Bangladesh. Kindly look at my comments to further elaborate in this sections.

Recommendations: Could be bolder, in light of your strong findings.

Reviewer #2: Title:

A suggestion is: Obstetricians’ perceptions of midwifery led care in Bangladesh – A qualitative study

Abstract:

Line 22: Is it midwifery that is a relatively new profession or is it the midwife as a separate profession?

Line 22-23: To better link the sentence to Obstetricians (line 23), the first sentence needs to include …. integration into the maternal healthcare system

Introduction

- Line 1: Suggest changing this reference to: The historical development of the midwifery profession in Bangladesh 2017 (Bogren, Begum, Erlandsson)

- Line 66: The sentence states that the introduction of midwifery in Bangladesh has been well documented. This is true and needs to be supported by a reference.

- To understand your findings, your introduction would benefit from a few sentences about how to become a midwife in Bangladesh. Several scientific papers have discussed this and proven that this education aligns with the ICM competencies for midwives.

- Line 78: The table of aims and objectives does not benefit the paper. The objectives could be presented as your research questions. Objective 2 is similar to the aim and can be removed.

Method

- Line 82-84: Please add a reference after a qualitative study.

- Line 86- 97: This paragraph would suit better in the strengths and limitation section.

- Please add a heading Setting, which will help the reader to understand in what context the study was taken place. Ex Number of births (Vaginal/CS), Number of obstetricians, midwives, nurses etc.

- Line 118: Please elaborate about how the participants were recruited (face-to face, email, advertisement etc).

- Line 121: Reconsider how your inclusion and exclusion criteria are presented, as it is presented now, your exclusion criteria is just the opposite of your inclusion criteria, which is redundant.

- Line 127: Please give some examples of your key questions to give the reader an understanding about what questions that were asked (even though your interview guide is attached)

- Line 134: I am a bit confused about the key informant. A Key informant is a person being interviewed and should therefore be part of the data collection. If I understand in correct, you present this person’s opinion in the discussion, when you discuss your results. See Example line 513. Why not using scientific articles to contextualize and address your results? What says that this person knows better than your participants and published scientific papers. Out of curiosity, was this person a midwife? Because that could give some nuanced tone to the discussion. However, there are so many publications out there made by experienced midwives with extensive knowledge of midwifery in Bangladesh.

- Line 150-157: The thematic analysis needs to be further clarified. How were the codes identified, how did you identify the categories, vs themes?

- What reference did you use for your thematic analysis? You mentioned that you made categories, usually in thematic analysis, themes and sub themes are used.

- To strengthen your analysis please provide an analysis matrix to exemplify the process.

- Please also include a table presenting themes and respective sub themes to provide an overview of your results.

Results

- Please use the term birth and childbirth instead of delivery. Women give birth

- Line 347: Please consider changing the theme to Lack of trust in midwives’ clinical education

- I am surprised to read that the obstetricians had opinions about the curriculum. when they were not fully up to the mark knowing the role of midwives,

Discussion

Line 443-445: Starting with the sentence a Key informant…..This information does not add to a scientific discussion. Suggest deleting,

Line 513: I would suggest referring to a scientific paper Instead of referring to Unfpa staff. There are several of published articles about midwifery education and educators in Bangladesh.

Conclusion

Do you have any suggestions on how to close the disconnect between obstetricians’ understanding of the role and scope of practice of Bangladeshi midwives and international definitions? Because, reading your findings made me quite sad after so many years of investments in building a midwife profession in Bangladesh.

**Do you want your identity to be public for this peer review?** For information about this choice, including consent withdrawal, please see our Privacy Policy

Reviewer #1: No

Reviewer #2: No

---

## [Decision Letter · Decision Letter 1]

5 Aug 2025

PGPH-D-24-02552R1

'Obstetricians' perceptions of midwifery-led care in Bangladesh - A qualitative study'

Dear Dr. Dafis,

Thank you for submitting your manuscript to PLOS Global Public Health. After careful consideration, we feel that it has merit but does not fully meet PLOS Global Public Health’s publication criteria as it currently stands. Therefore, we invite you to submit a revised version of the manuscript that addresses the points raised during the review process.

EDITOR: Please carefully read and address the reviewer's comments and provide a point to point response to reviewer's comments.

We look forward to receiving your revised manuscript.

Kind regards,

Dr Tanmay Bagade, Ph.D., MS (O&G), MPH, MHM

Academic Editor

Journal Requirements:

Reviewers' comments:

Reviewer's Responses to Questions

**Comments to the Author**

Reviewer #1: All comments have been addressed

Reviewer #2: All comments have been addressed

publication criteria?

Reviewer #1: Partly

Reviewer #2: Yes

3. Has the statistical analysis been performed appropriately and rigorously?

Reviewer #1: N/A

Reviewer #2: N/A

4. Have the authors made all data underlying the findings in their manuscript fully available (please refer to the Data Availability Statement at the start of the manuscript PDF file)?

Reviewer #1: Yes

Reviewer #2: No

5. Is the manuscript presented in an intelligible fashion and written in standard English?

Reviewer #1: No

Reviewer #2: Yes

Reviewer #1: Thank you for addressing my comments. I have a few minor suggestions you may consider:

1. There are several grammatical errors, such as inappropriate use of commas and apostrophes. There is a mix of present and past tenses in your methods and findings sections, both are traditionally written in past tense. This could be corrected by getting the manuscript reviewed by an English language expert.

2. What is CIPRB (line 131)? Requesting you to include full forms of all acronyms at the time of first use.

3. Authors describe trustworthiness improving measures in detail (Lines 184-196). How about the other three measures: confirmability, credibility and transferability?

Reviewer #2: Thank you for addressing my feedback. I appreciate that you have removed the comments from the Key informants from the Discussion. Please reconsider if you need the section about the same in the method section. I would suggest to remove it, as this is not presented in the analysis.

**Do you want your identity to be public for this peer review?** For information about this choice, including consent withdrawal, please see our Privacy Policy

Reviewer #1: No

Reviewer #2: No

---

## [Editor Report · Decision Letter 2]

6 Nov 2025

'Obstetricians' perceptions of midwifery-led care in Bangladesh - A qualitative study'

PGPH-D-24-02552R2

Dear Miss Dafis,

We are pleased to inform you that your manuscript ''Obstetricians' perceptions of midwifery-led care in Bangladesh - A qualitative study'' has been provisionally accepted for publication in PLOS Global Public Health.

Best regards,

Dr Tanmay Bagade

Academic Editor